# Overweight and Obesity: Its Impact on Foot Type, Flexibility, Foot Strength, Plantar Pressure and Stability in Children from 5 to 10 Years of Age: Descriptive Observational Study

**DOI:** 10.3390/children10040696

**Published:** 2023-04-07

**Authors:** Cristina Molina-García, José Daniel Jiménez-García, Daniel Velázquez-Díaz, Laura Ramos-Petersen, Andrés López-del-Amo-Lorente, Carlos Martínez-Sebastián, Francisco Álvarez-Salvago

**Affiliations:** 1Health Sciences PhD Program, Universidad Católica de Murcia UCAM, Campus de los Jerónimos n°135, 30107 Murcia, Spain; 2Department of Health Sciences, Faculty of Health Sciences, University of Jaén, 23071 Jaen, Spain; 3AdventHealth Research Institute, Neuroscience Institute, Orlando, FL 32803, USA; 4ExPhy Research Group, Department of Physical Education, Faculty of Education Sciences, University of Cadiz, 11519 Cadiz, Spain; 5Department of Nursing and Podiatry, University of Malaga, 29071 Malaga, Spain; 6Department of Physiotherapy, Faculty of Health Sciences, European University of Valencia, 46010 Valencia, Spain

**Keywords:** pediatric obesity, children, plantar pressure, static stability

## Abstract

Background: Overweight (OW) and childhood obesity (OB) may cause foot problems and affect one’s ability to perform physical activities. The study aimed to analyze the differences in descriptive characteristics, foot type, laxity, foot strength, and baropodometric variables by body mass status and age groups in children and, secondly, to analyze the associations of the BMI with different physical variables by age groups in children. Methods: A descriptive observational study involving 196 children aged 5–10 years was conducted. The variables used were: type of foot, flexibility, foot strength and baropodometric analysis of plantar pressures, and stability by pressure platform. Results: Most of the foot strength variables showed significant differences between the normal weight (NW), OW and OB groups in children aged between 5 and 8. The OW and OB groups showed the highest level of foot strength. In addition, the linear regression analyses showed, in children aged 5 to 8 years, a positive association between BMI and foot strength (the higher the BMI, the greater the strength) and negative association between BMI and stability (lower BMI, greater instability). Conclusions: Children from 5 to 8 years of age with OW and OB show greater levels of foot strength, and OW and OB children from 7 to 8 years are more stable in terms of static stabilometrics. Furthermore, between 5 and 8 years, having OW and OB implies having more strength and static stability.

## 1. Introduction

The World Health Organization (WHO) defines overweight (OW) and obesity (OB) as an abnormal accumulation of fat that represents a health risk [1]. It is also considered as “one of the most serious public health challenges of the 21st century” due to the fact that childhood OB continues to rise and more frequently occurs at younger ages with more serious health consequences associated with the early onset of OB [2,3]. According to the WHO, the prevalence of OW and OB in children and adolescents aged between 5 and 19 increased during the past years, rising from 4% of that population in 1975 to 18% in 2016 [4]. The World Obesity Federation has already stated that in 2030, 254 million children and adolescents will suffer from OB [5]. Interestingly, the vast majority of OW or obese children live in developing countries, where the rate of increase has been more than 30% higher than in developed countries in 2022 [1].

Considering previous literature, some studies have already highlighted how children who have OW and OB are more likely to suffer from several clinical comorbidities such as diabetes or metabolic syndrome [6], while others have already remarked how there is an association between increased risk of injury and childhood OB and gait, plantar pressures and stability [7,8,9], these being the main factors related to pain in the feet and lower limbs in children [10]. Notwithstanding, there is still controversy about the level of association between OW and OB and gait disturbance in children. In this sense, some studies have reported that children with OW and OB have weaker stability, a flatter foot pattern and a larger axis of the foot than normal weight (NW) children, which seems to impact plantar pressures [10,11,12,13], provoke pain and affect their quality of life [14,15]. Other studies, on the contrary, described no relationship between the OW and OB and foot pronation [16,17]. Therefore, and bearing in mind the lack of consensus and evidence, more studies are still needed to shed more light on this subject, which could be very useful for the clinical management of these young patients.

At the clinical level, the impact of gait is of great importance due to the fact that there is a large inverse correlation between physical activity level and plantar pressure [18]. This is mainly because deformities of the musculoskeletal system can be caused by an increase in pressure as a result of an increase in body mass index (BMI) [10]. Thus, by increasing plantar pressure, pain would increase and, consequently, the ability to perform physical activity would be limited, which could result in an impairment of children’s quality of life [14,15]. In fact, it is known that suffering from alterations of this type limits the motivation of children to perform physical activities, which would further aggravate the problem of inactivity and suffering from OW and OB [19]. Considering all of the above, as the main clinical implication, all risk factors related to the onset of pain must be recognized. Signs such as excessive pressure or altered stability, should be recognized to prevent pain and complications in the short and long term, that is, to work from early prevention.

Therefore, this study aims to analyze the differences in descriptive characteristics, foot type, laxity, foot strength and baropodometric variables by body mass status and age groups in children and, secondly, to analyze the associations of BMI with variables foot type, laxity, foot strength and baropodometric variables by age groups in children.

## 2. Materials and Methods

### 2.1. Participants

One hundred and ninety-six children (78 males and 118 females) aged between 5 and 10 years were recruited for this descriptive observational study. This age range was chosen because it is the age at which the Foot Posture Index (FPI) is validated [20]. The study was approved by the Ethics Committee of the Catholic University of San Antonio de Murcia (Spain) (Code: CE022205). For the realization of the study, the Strengthening Reporting of Observational Studies in Epidemiology (STROBE) guidelines have been followed [21]. This study was performed in line with the principles of the Declaration of Helsinki [22].

The selection criteria of the sample were children aged 5–10 years, who do not have foot pain and who had the consent of the parents/guardians. Parents/guardians were previously informed about the study, completed a questionnaire and signed their consent to confirm their children’s participation. Children who had any of the following conditions were excluded from the study: recent damage to lower limbs; congenital structural alterations affecting distal areas of the ankle joint, as well as those cases with pathological flatfoot caused by cerebral palsy; surgical treatments in the foot or lower extremity; or genetic and neurological or muscular pathology.

All children were evaluated between February and June 2022 in a primary school in the region of Murcia (Spain). For 5 months, all participating children completed all the assessments in one morning on the same schedule. Demographic and anthropometric data were collected from all children prior to the investigation. Children were assigned a specific number to maintain confidentiality. To examine them, they were asked to be barefoot and in light clothing (t-shirts and shorts) and were individually evaluated by two expert clinicians at the same time. If these two clinicians disagreed during measurements of the same child, a third clinician decided which of the two values was more accurate. Similarly, another clinician who was not involved in the assessments was in charge of analyzing the data. 

Before starting the test, each test was demonstrated and explained to every child. In this sense, each item of each test had to be performed three times, and the measurements obtained were averaged. Children received standard verbal encouragement and support throughout the whole testing procedure. When a child made a procedural error, the instructions and demonstrations were repeated and the child was allowed to try again; each child was allowed to fail a maximum of 5 times. All children completed all the measurements correctly.

### 2.2. Measurement of the Variables

#### 2.2.1. Anthropometric Measures

Height was measured with a calibrated portable SECO 7710 m, with a bubble level fixed to the arm for greater accuracy, while weight was measured with Digital Pegasus Scales, with a margin of error of 0.05 kg and keeping subjects with as little clothing as possible (shirt and shorts).

To establish cut-off points for specific BMI by sex and age, cut-off points established in previous bibliography were considered [23,24], children were categorized as normal weight “NW”, overweight “OW” or obese “OB”.

#### 2.2.2. Type of Foot, Laxity and Foot Strength

To find out what type of foot each child had, the evaluation of each foot was carried out by measuring the Foot Posture Index (FPI-6) with the subjects standing barefoot, in a relaxed position, on a 50 cm bench to facilitate visual and manual inspection. The FPI-6 rates 6 aspects of foot anatomy in the 3 planes of the foot. The FPI-6 takes into account the posture of the hindfoot, midfoot and forefoot. The FPI-6 provides a total value from −12 points (highly supinated) to +12 points (highly pronated). Interobserver reliability for FPI-6 in the pediatric population has reached a consistent weighted Kappa value (Kw = 0.86) in a sample of children aged 5 to 16 years of age [25].

To recognize whether the children had joint hypermobility (JH) or hyperlaxity, two scales and one test were also used: the Beighton Scale [26], the Lower Limb Assessment Score (LLAS) [27], and the Ankle Lunge Test [28]. For both scales, goniometry was used, which is a valid instrument to measure generalized joint mobility in school-age children [29].

The Beighton scale is used to observe if the child presents JH at a general level, that is, in the wrist, the metacarpophalangeal joint of the fifth metacarpal, in the elbow, in the knee (all bilateral and without weight bearing) and in the lumbosacral spine. The Beighton scale has a score of 9 points, so the usual arbitrary cutoff of 5/9 or higher indicates that the child has JH. This scale has shown to be reliable, with a Kw 0.81 [26]. 

The LLAS measures JH, but of the lower extremity. The hip, knee, ankle, subtalar joint, midtarsal joint and metatarsophalangeal joint were assessed. On the LLAS scale, each limb produces a final score of 12 points, so a score of 7/12 or higher indicates JH. The LLAS has shown to be reliable, with an intraclass correlation coefficient (ICC) of 0.84 [27]. 

The Ankle Lunge Test assesses the range of weight-bearing ankle dorsiflexion with the knee flexed. To quantify the Ankle Lunge Test, a digital inclinometer (Smart Tool^TM^) was used, which was applied to the anterior surface of the tibia to measure ankle dorsiflexion. This test has shown to be reliable, with an intra-assessed ICC of 0.98 and an inter-assessed ICC of 0.97 [28].

Isometric muscle strength was quantified using the Lafayette Instrument Company Hand Dynamometer, Model 01160, Lafayette, Indiana, U.S.A. The device was calibrated at the factory, according to the manufacturer’s data, at a sensitivity of 0.1 kg and a range of 0.0 to 199.9 kg. Each child was placed in a long sitting position (hips flexed and knees extended) on an examination table with a backrest. Isometric foot inversion and eversion muscle strength, and ankle plantarflexion and dorsiflexion was measured according to a standardized procedure [30]. This measurement has shown good intra-rater (ICC = 0.92–0.97) and inter-rater (ICC = 0.80–0.95) reliability [31].

### 2.3. Baropodometry

The baropodometric analysis was performed with the RSscan Footscan^®^ 9 platform, with dimensions of 578 mm × 418 mm × 12 mm. The platform contains 4096 sensors (arranged in a 64 × 64 matrix), the dimensions of the sensors are 7.62 mm × 5.08 mm and the active area is 488 mm × 325 mm. The precision range is 1–127 N/cm^2^ and the data acquisition frequency is 500 Hz with a 10-bit resolution. 

Following the manufacturer’s manual, the platform was calibrated before each session. Three baropodometric measurements in an orthostatic position and three stabilometric measurements in an orthostatic position with eyes open for 60 s were taken for each child; a minute of rest was left between each measurement. Children were asked to stand on the platform, with their own Fick angle, arms along the body, feet at the same height and facing forward towards a fixed point that was placed at eye level at a distance of 3.8 m. Before data collection, children were allowed to familiarize themselves with the platform until they were confidently able to perform it.

The parameters considered were the % of pressure distribution in left and right leg, forefoot and rearfoot, left (C1) and right (C2) forefoot and left (C3) and right (C4) rearfoot for both static and stabilometry. In addition, in the stabilometric measurement, the following parameters were measured:Position (minimum–maximum *x*-*y* axis): the current, minimum and maximum position in millimeter for the x- and y-coordinate;Range (interval–average x-y): the spread between the minimum and maximum position in millimeters for the x- and y-coordinate;Travelled distance: the length of the center of pressure line in millimeters;Ellipse area: the area of the calculated center of pressure ellipse in square millimeters;Principal–second axis ellipse: Length of the major–minor axis of the ellipse of the center of pressure of the left and right foot, measured in millimeters.

The reliability of the Footscan^®^ system (RSscan International, Olen, Belgium) has been demonstrated by different investigations with ICC values from good to excellent for the intra- and inter-evaluator scores (ICC 0.81–0.86 and ICC 0.87–0.95, respectively) on plantar pressure variables [32].

### 2.4. Statistical Analysis

All variables were checked for normality using both graphical and statistical procedures. Differences in descriptive characteristics, type of foot, laxity, foot strength and baropodometric characteristics of the overall sample by age and body mass status were examined applying the t-test. For that, three age groups were created: (1) 5–6 years; (2) 7–8 years; and (3) 9–10 years. Moreover, two BMI groups were also created: (1) children with NW and (2) children with OW and OB.

Then, linear regression analyses were performed to analyze the association of BMI with type of foot, laxity, foot strength and baropodometric variables across all three age groups. Previously, sex interaction was analyzed by including the interaction terms in the code of regression analyses. Since there was no sex interaction, the sample was segmented by age and BMI. In addition, the collinearity of the regression models was calculated using command .vif, which did not show independent variables with a coefficient > 10. Finally, for each regression model, the normality analyses were recalculated for the residuals for the models.

All analyses were performed using the STATA software for Windows version 13.0. The level of significance was set at *p* < 0.05.

## 3. Results

### 3.1. Descriptive Characteristics, Type of Foot, Laxity, Foot Strength and Baropodometric Characteristics of the Sample by Age and BMI in Children

The characteristics of the sample by age and body mass status are shown in Table 1. In terms of descriptive characteristics, there were no significant differences between the NW group and the OW and OB groups for the three age groups with respect to age and height (all *p* > 0.005). However, there were significant differences between the NW group and the OW and OB groups in all three age groups for weight and BMI (all < 0.005).

Then, type of foot and laxity were analyzed; however, there were no significant differences between NW groups and OW and OB groups in all three age groups for any of the variables (all *p* > 0.05). In relation to foot strength variables, most variables of eversion, inversion and plantar flexion and dorsiflexion strength showed significant differences between the NW group and the OW and OB groups in the 5–6 and 7–8 years groups (*p* < 0.005). However, there were no significant differences between the NW group and the OW and OB group in the 9 to 10 years group for foot strength variables (*p* > 0.005).

As for static variables, there were no significant differences between the NW group and the OW and OB groups in all three age groups for any of the variables (all *p* > 0.05). This same trend was observed with the analysis of the stabilometric variables, where there were no significant differences between the NW group and the OW and OB groups in all three age groups for none of the variables (all *p* > 0.05), except for the interval and distance traveled in the 7–8 years group (both *p* < 0.05).

### 3.2. Associations of BMI with Type of Foot, Laxity, Foot Strength and Baropodometric Variables by Age Groups in Children

The regression analyses of the BMI with type of foot, laxity, foot strength and baropodometric characteristics by age group are shown in Table 2. Regarding type of foot and laxity, there were no significant associations of BMI with the variables analyzed (all *p* > 0.05), except for the Beighton Scale in children 7–8 years (*p* < 0.049). In relation to foot strength variables, all variables showed positive significant associations of BMI with eversion, inversion and plantar flexion and dorsiflexion strength (all *p* < 0.05), except for right eversion and left dorsiflexion in children 5–6 years. In children 7–8 years, BMI was positively associated with left and right inversion strength (all *p* < 0.05). However, there were no significant associations of BMI with foot strength variables in children 9 to 10 years (all *p* > 0.05).

As for the associations of BMI with static variables, there were no significant associations of BMI with any of the static variables in all three age groups (all *p* > 0.05), except for C4 static in children 7–8 years (*p* = 0.043). Then, the associations of BMI with stabilometric variables were performed. There were no significant associations of BMI with any of the stabilometric variables in all three age groups (all *p* > 0.05), except for left–right difference in children 9 to 10 years (*p* = 0.032). In relation to gravity center, BMI was negatively associated with interval x, distance traveled, and secondary axis ellipse in children 5–6 years (all *p* < 0.05). Finally, BMI was negatively associated with interval x and y, maximum *y*-axis and principal axis ellipse in children 7–8 years (all *p* < 0.05). However, there were no significant associations of BMI with any of the gravity center variables in children 9–10 years (all *p* > 0.05).

## 4. Discussion

The purpose of this study was to analyze the differences in descriptive characteristics, foot type, laxity, foot strength and baropodometric variables by body mass status and age groups in children and, secondly, to analyze the associations of BMI with foot type, laxity, foot strength and baropodometric variables by age groups in children.

The main findings of the present work revealed that most foot strength variables showed significant differences between the NW groups and the OW and OB groups in children 5–6 and 7–8 years, OW and OB children having a higher level. Moreover, some stabilometric variables showed significant differences between the NW group and the OW and OB group in children 7–8 years. Then, linear regression analyses showed positive associations of BMI with most of the foot strength variables in children 5–6 and 7–8 years, as well as negative associations with the gravity center variables.

Considering our results, it can be observed that OW and OB children between 5 and 8 years have significantly higher levels of foot strength compared to NW children and that NW children between 7 and 8 years show worse stabilometric values compared to children with OW and OB. On the one hand, and despite the limited evidence, only a few articles have shown to date how the foot type, strength and flexibility can influence foot structure, pressure distribution and other possible musculoskeletal disorders [10,13,14]. In this regard, our results reflect how children with OW and OB had more foot isometric strength. No previous studies have shown a relationship between OW and OB and isometric strength of the foot. However, a previous study showed a relationship between OW and OB and isometric strength of the hands [33]. Thus, in our humble opinion, we believe this is the first study to assess the impact of isometric foot strength in children with OW and OB. On the other hand, and although the evidence so far supports the fact that children with OW and OB are less stable when walking [7], it is possible that this dissimilarity was caused by the same fact mentioned by Kjölhede et al. (2014), in which they concluded that children with NW tend to be more restless than children with OW and OB [34]. Therefore, considering that, in our study, we analyzed the pressures in static/stabilometry and not in dynamic motion could explain why NW children showed worse stabilometry.

This study also explored in the regression analysis the association between OW and OB on the other dependent variables. Firstly, it is important to remark that there is no significant association between type of foot (FPI-6) and OW and OB. In this sense, these results are in contrast and in line with those shown by previous literature, as some authors have concluded that there was a correlation between flatter feet and children with OW and OB [35,36], while others have stated that there is no relationship between increased BMI in children and having “flatter” feet [37,38,39]. In this sense, we believe that the controversy among these studies investigating the relationship between BMI and OW and OB could be the method of grading the foot. Therefore, future studies should unify the method of evaluating flatfoot to facilitate the comparison of results and to be able to draw more accurate and precise conclusions.

When it comes to JH, previous evidence showed discrepancies because some studies have shown that children with OW and OB have a stronger relationship with JH [40], while others confirm that JH is more prevalent in underweight children [41]. In this sense, our results suggest that having a higher BMI is associated with having more JH overall from 7 to 8 years of age. Hence, more studies are needed to corroborate this association, since JH is a risk factor for musculoskeletal pain during adolescence [40].

As far as we know at present, there are no studies that relate BMI to ankle muscle strength and its possible involvement with excess plantar pressures, gait biomechanics and musculoskeletal alterations in the lower limbs in children. This issue has only been addressed in the adult population, where it has been observed that OW and OB decrease ankle muscle strength and quality of life [42,43]. In our study, we have observed how an increase in BMI is directly related to a greater isometric strength of ankle movements (inversion, eversion, dorsiflexion and plantar flexion) mainly in younger children (5–6 years of age). At this point, even if OB children show more strength, it is still of clinical alarm since this can translate into joint overload and having more strength does not mean that they execute movements more correctly, a fact that has already been mentioned in previous research [33,44,45,46].

Hereinafter, the results of the present study are consistent with previous research, where it was reported that an increase in BMI is related to alterations in static plantar pressures and stabilometry in children [7,8,47,48]. In this way, our regression analyses showed in children 7–8 years of age that the pressure in the right heel is significantly higher. In this sense, studies such as Bittar et al. [49] and Feka et al. [48] showed that BMI is associated with greater pressure in the hindfoot. Additionally, and regarding the difference in pressure between the left and right leg, we also found that the BMI mainly influences children of the age of 9–10 years, who tend to receive more pressure in the right foot. In this sense, this result is in line and in contrast with previous research, since some studies mention that there is more support in the right foot [48,50,51], while Bittar el al. [49] mention that there is more support in the left foot. Therefore, more studies are still needed to clarify this fact in children because an asymmetrical distribution of loads could lead to asymmetrical growth of the limbs or overloads, leading to postural deformities.

Finally, we could also observe a relationship between having a higher BMI and presenting better static stability in children from 5 to 8 years of age; that is, they had fewer oscillations. However, it is important to remark that this better static stability due to higher BMI values could also be translated into a worse capacity to compensate for the overload that their feet receive due to excess weight. This clinical reasoning is built on the results of previous studies that have already highlighted the impact of OW and OB on stability [7,11,52], remarking how an excessive BMI leads to mechanical overexertion which cannot be compensated for by the musculoskeletal system [53]. The basis of most movements is due to balance control [7], so if this control is affected, it would also affect the daily living activities of children with OW and OB. Hence, these findings seem to confirm that OW and OB negatively impact the normal musculoskeletal development of children’s feet compared to children with NW. In this way, we also dare to speculate that, in turn, this could have a negative impact on their quality of life and global health status.

Although there are a wide variety of plantar pressure measurement systems [54,55,56], such as the use of instrumented insoles, the pressure platform was used because in children the size of the foot varies greatly from one child to another, even more in an age range as wide as 5 to 10 years of age. Perhaps if the instrument templates had been used instead of the pressure platform, the data would have been more accurate.

This study has several limitations that deserve attention. First, the values used as cut-off points to divide children as normal weight “NW”, overweight “OW” or obese “OB” has been previously used and accepted [23,24], although other cut-off values could have modified our results. Secondly, although the RSscan Footscan^®^ 9 pressure platform has demonstrated good intra-rater and inter-rater reliability [32], it only measures forces perpendicular to the ground, not taking into account forces on other planes. Thirdly, it should take into account that the age range of our study was from 5– to 10 years of age, so direct comparisons with other studies could be difficult due to other possible age ranges. Finally, the data collected through the pressure platform are static; hence, we cannot just infer that static positioning will directly impact dynamic movements.

Despite these cited limitations, this study has several strengths. First, it comprised a wide age range of children. Secondly, is the first study to assess foot type, strength and flexibility in the same sample of obese children. Thirdly, the measuring instruments implemented in our study are widely used in both clinical practice and research, which, together with the data obtained and taking into account the increasing rate of childhood OB [1,2,3], our findings may have important clinical and public health implications.

The clinical implications of the findings presented in this study imply that signs such as excessive pressure, impaired stability or increased foot strength must be recognized to prevent future pain and possible short- and long-term complications. OW and OB prophylaxis, which is becoming more frequent every day, as well as early diagnosis of musculoskeletal deformities, will have long-term effects on the general health status of children. An alteration in the feet and all that this implies (strength, flexibility, pressure, stability) can have consequences such as decreased physical activity, aggravating the OW and OB problem. Children with OW and OB should be managed by a multidisciplinary team, which should be made up of psychologists, nutritionists, pediatricians, rehabilitators, podiatrists and physiotherapists.

## 5. Conclusions

Children from 5–8 years of age with OW and OB show greater levels of foot strength and also how OW and OB children from 7–8 years are more stable in static stabilometrics. Furthermore, the linear regression analyses showed how, between 5 and 8 years, having OW and OB implies having more strength and static stability. This should not be translated as a positive aspect for health in this population. Considering the scarcity of studies, that OB rates continue to grow and that having greater strength and that stability as a consequence of a higher BMI is not beneficial to health, more studies are still needed in this regard in order to provide a more adequate management of the consequences of OB in children.

## Figures and Tables

**Table 1 children-10-00696-t001:** Descriptive characteristics, type of foot, laxity, foot strength and baropodometric characteristics of the sample by age and body mass status in children.

	5 to 6 Years				7 to 8 Years				9 to 10 Years			
Variables	Total n = 79 M/F	NW n = 55 M/F	OW and OB n = 24 M/F	*p* ^a^	Total n = 67 M/F	NW n = 45 M/F	OW and OB n = 22 M/F	*p* ^a^	Total n = 50 M/F	NW n = 29 M/F	OW and OB n = 21 M/F	*p* ^a^
**Physical characteristics**												
Age (years)	6.14 ± 0.48	6.13 ± 0.48	6.17 ± 0.46	0.742	7.98 ± 0.62	7.92 ± 0.58	8.11 ± 0.67	0.221	9.51 ± 0.29	9.52 ± 0.29	9.51 ± 0.40	0.944
Weight (kg)	23.10 ± 4.47	21.22 ± 2.72	27.42 ± 4.75	**<0.001**	29.48 ± 7.18	25.62 ± 3.08	37.38 ± 6.68	**<0.001**	37.30 ± 7.36	33.25 ± 4.04	42.89 ± 7.96	**<0.001**
Height (cm)	1.17 ± 0.05	1.17 ± 0.05	1.18 ± 0.05	0.309	1.28 ± 0.07	1.27 ± 0.06	1.30 ± 0.06	0.160	1.39 ± 0.07	1.39 ± 0.06	1.38 ± 0.07	0.512
BMI (kg·m^−2^)	16.77 ± 2.27	15.54 ± 1.02	19.60 ± 2.28	**<0.001**	17.88 ± 3.48	15.95 ± 1.07	21.83 ± 3.39	**<0.001**	19.30 ± 3.53	17.06 ± 1.26	22.38 ± 3.53	**<0.001**
Gender, *n* (%)	39(49)/40(51)	30 (55)/25(45)	9(38)/15(62)	**0.857 ^b^**	24(36)/43(64)	16 (36)/29(64)	8(36)/14(64)	**0.232 ^b^**	15(30)/35(70)	8(28)/21(72)	7(33)/14(67)	**0.178 ^b^**
**Type of foot, laxity and foot strength**												
FPI total (Score)	7.76 ± 5.56	7.42 ± 5.31	8.54 ± 5.98	0.407	7.39 ± 5.58	7.53 ± 5.89	7.09 ± 4.99	0.763	8.34 ± 5.24	9.00 ± 0.97	7.42 ± 5.70	0.318
Lunge test (^o^)	106.7 ± 10.5	108.2 ± 10.4	103.4 ± 10.3	0.065	97.62 ± 13.12	99.00 ± 13.62	94.81 ± 11.85	0.223	95.40 ± 12.24	95.14 ± 13.26	95.76 ± 10.97	0.860
Beighton (Score)	3.49 ± 2.98	3.71 ± 3.14	3.00 ± 2.59	0.334	3.10 ± 3.07	2.67 ± 3.02	4.00 ± 3.05	0.095	2.34 ± 2.73	2.79 ± 2.82	1.71 ± 2.53	0.170
R LLAS (Score)	6.64 ± 3.49	7.05 ± 3.56	5.71 ± 3.21	0.116	5.43 ± 3.54	5.20 ± 3.62	5.91 ± 3.39	0.445	4.30 ± 3.18	4.86 ± 3.40	3.52 ± 2.77	0.145
L LLAS (Score)	6.51 ± 3.55	6.85 ± 3.61	5.71 ± 3.35	0.189	5.42 ± 3.48	5.20 ± 3.58	5.86 ± 3.31	0.468	4.30 ± 3.33	4.83 ± 3.50	3.57 ± 3.01	0.190
R eversion (N)	6.62 ± 2.90	6.39 ± 3.37	7.14 ± 1.21	0.294	7.52 ± 1.88	7.10 ± 1.89	8.37 ± 2.03	**0.013**	11.84 ± 2.37	12.02 ± 2.34	11.59 ± 2.43	0.536
L eversion (N)	6.00 ± 1.44	5.70 ± 1.32	6.67 ± 1.48	**0.005**	7.22 ± 2.47	6.99 ± 2.68	7.68 ± 1.95	0.290	11.28 ± 2.09	11.33 ± 1.73	11.22 ± 2.56	0.877
R inversion (N)	7.38 ± 1.34	7.10 ± 1.16	8.05 ± 1.51	**0.003**	8.67 ± 1.58	8.34 ± 1.56	9.33 ± 1.44	**0.014**	12.33 ± 2.02	12.42 ± 2.07	12.20 ± 1.99	0.717
L inversion (N)	6.67 ± 1.37	6.47 ± 1.29	7.12 ± 1.46	0.054	7.75 ± 1.83	7.42 ± 1.75	8.39 ± 1.88	**0.041**	11.66 ± 1.79	11.70 ± 1.65	11.60 ± 2.01	0.841
R. plantarflexion (N)	10.59 ± 2.62	10.04 ± 2.03	2.63 ± 3.37	**0.004**	14.28 ± 3.88	13.85 ± 3.68	15.15 ± 4.20	0.199	26.06 ± 5.77	25.89 ± 5.50	26.30 ± 6.27	0.810
L plantarflexion (N)	10.21 ± 2.73	9.78 ± 2.22	11.19 ± 3.50	**0.034**	14.52 ± 5.84	13.56 ± 4.19	16.46 ± 8.03	0.055	25.17 ± 5.66	24.78 ± 5.17	25.70 ± 6.36	0.574
R dorsiflexion (N)	6.67 ± 1.18	6.45 ± 1.02	7.18 ± 1.38	**0.010**	7.53 ± 1.59	7.25 ± 1.46	8.11 ± 1.70	**0.036**	10.45 ± 1.39	10.55 ± 1.21	10.33 ± 1.63	0.590
L dorsiflexion (N)	6.71 ± 2.16	6.62 ± 2.38	6.94 ± 1.56	0.544	7.21 ± 1.71	6.89 ± 1.50	7.87 ± 1.94	**0.025**	10.17 ± 1.56	10.22 ± 1.16	10.09 ± 1.83	0.755
**Static variables (%)**												
R-L difference static	7.78 ± 6.32	7.77 ± 6.68	7.80 ± 5.55	0.981	7.55 ± 5.63	7.22 ± 5.63	8.23 ± 5.71	0.497	7.32 ± 6.49	6.80 ± 4.22	8.05 ± 6.49	0.415
Forefoot static	41.40 ± 8.47	41.40 ± 8.96	38.76 ± 6.65	0.067	42.27 ± 8.26	42.93 ± 9.03	40.91 ± 6.36	0.351	43.86 ± 7.80	44.16 ± 8.50	43.44 ± 6.90	0.749
Rearfoot static	58.56 ± 8.46	57.39 ± 8.95	61.23 ± 6.66	0.063	57.72 ± 8.25	57.05 ± 9.02	59.08 ± 6.36	0.346	56.14 ± 7.80	55.83 ± 8.50	56.56 ± 6.90	0.749
C1 static	22.01 ± 7.03	22.76 ± 7.67	20.30 ± 4.98	0.152	21.74 ± 4.91	22.32 ± 5.35	20.56 ± 3.70	0.171	22.52 ± 5.00	22.92 ± 5.45	21.96 ± 4.36	0.508
C2 static	19.84 ± 4.03	20.33 ± 4.13	18.72 ± 4.57	0.126	20.55 ± 4.84	20.65 ± 4.88	20.34 ± 4.87	0.811	21.35 ± 4.45	21.24 ± 4.76	21.49 ± 4.09	0.845
C3 static	30.27 ± 6.17	29.58 ± 6.66	31.86 ± 4.60	0.131	29.86 ± 5.35	30.05 ± 5.82	29.46 ± 4.35	0.675	29.02 ± 5.16	29.04 ± 4.52	29.00 ± 6.05	0.977
C4 static	28.33 ± 5.58	27.87 ± 5.89	29.38 ± 4.73	0.274	27.85 ± 5.99	26.99 ± 6.26	29.62 ± 5.06	0.092	27.12 ± 6.79	26.78 ± 7.24	27.57 ± 6.27	0.689
**Stabilometric variables (static) (%)**												
R-L difference stabilometric	9.25 ± 7.18	9.30 ± 7.57	9.14 ± 6.33	0.929	7.30 ± 5.92	7.38 ± 5.52	7.13 ± 5.93	0.872	9.00 ± 8.12	9.28 ± 6.29	8.60 ± 10.28	0.773
Forefoot stabilometric	39.07 ± 6.44	39.84 ± 7.21	37.30 ± 3.73	0.107	40.93 ± 7.74	40.82 ± 8.92	41.13 ± 4.61	0.880	42.91 ± 8.12	44.04 ± 8.27	41.35 ± 5.28	0.198
Rearfoot stabilometric	60.82 ± 6.49	60.16 ± 7.21	62.35 ± 4.16	0.169	59.07 ± 7.73	59.17 ± 8.92	58.86 ± 7.74	0.880	57.09 ± 7.24	55.96 ± 8.27	58.64 ± 5.31	0.199
C1 stabilometric	20.49 ± 3.47	20.93 ± 5.14	19.48 ± 3.47	0.208	21.14 ± 4.40	21.19 ± 4.89	21.05 ± 3.26	0.912	22.01 ± 4.38	22.72 ± 4.72	21.01 ± 3.74	0.175
C2 stabilometric	18.62 ± 3.61	18.91 ± 3.79	17.96 ± 3.12	0.286	19.77 ± 4.84	19.62 ± 5.41	20.08 ± 3.51	0.724	20.92 ± 4.31	21.33 ± 4.81	20.36 ± 3.55	0.438
C3 stabilometric	19.84 ± 4.13	20.33 ± 4.13	18.72 ± 4.57	0.126	20.55 ± 4.84	20.64 ± 4.88	20.34 ± 4.88	0.811	21.35 ± 4.45	21.24 ± 4.76	21.49 ± 4.09	0.845
C4 stabilometric	32.47 ± 5.11	32.13 ± 5.77	33.22 ± 3.11	0.386	30.40 ± 5.37	30.77 ± 5.84	29.63 ± 4.27	0.568	27.28 ± 6.06	26.87 ± 6.70	27.83 ± 5.16	0.583
**Stabilometric variables (gravity center) (mm)**												
Minimum *x*-axis	−0.72 ± 5.64	−1.05 ± 6.27	0.04 ± 3.86	0.431	−0.40 ± 4.18	−0.40 ± 4.36	−0.41 ± 3.88	0.993	1.16 ± 2.96	0.97 ± 3.04	1.43 ± 2.89	0.590
Minimum *y*-axis	−7.82 ± 4.60	−8.16 ± 4.97	−7.04 ± 3.59	0.322	−6.52 ± 3.54	−6.77 ± 3.83	−6.00 ± 2.89	0.403	−7.42 ± 4.49	−7.93 ± 4.49	−6.71 ± 4.20	0.337
Maximum *x*-axis	7.13 ± 4.01	7.36 ± 4.01	6.58 ± 4.04	0.429	6.87 ± 4.43	7.49 ± 4.78	5.59 ± 3.33	0.099	7.64 ± 4.02	7.69 ± 3.92	7.57 ± 4.25	0.919
Maximum *y*-axis	1.58 ± 4.40	1.65 ± 4.76	1.42 ± 3.51	0.826	1.57 ± 3.56	2.04 ± 3.80	0.59 ± 2.82	0.116	1.00 ± 4.44	0.21 ± 4.49	2.10 ± 5.40	0.140
Interval x	7.78 ± 5.45	8.35 ± 5.91	6.50 ± 4.00	0.167	7.33 ± 4.28	7.98 ± 4.74	6.00 ± 2.76	0.075	6.66 ± 3.29	7.07 ± 3.60	6.09 ± 2.79	0.306
Interval y	9.49 ± 4.93	9.85 ± 5.45	8.67 ± 3.42	0.327	8.09 ± 4.28	8.82 ± 4.13	6.59 ± 2.75	**0.036**	8.48 ± 3.89	8.21 ± 3.93	8.86 ± 3.90	0.565
Average x	3.76 ± 5.02	3.80 ± 5.53	3.67 ± 3.66	0.914	3.32 ± 4.03	3.64 ± 4.21	2.68 ± 3.63	0.362	4.50 ± 3.51	4.38 ± 3.58	4.66 ± 3.50	0.778
Average y	−2.94 ± 3.53	−3.07 ± 3.75	−2.67 ± 3.03	0.641	−2.42 ± 2.86	−2.24 ± 3.25	−2.77 ± 2.86	0.519	−3.14 ± 4.06	−3.82 ± 3.84	−2.19 ± 4.25	0.161
Distance traveled	50.19 ± 29.34	54.11 ± 31.87	41.21 ± 20.35	0.072	42.65 ± 29.33	47.71 ± 33.30	32.32 ± 14.66	**0.042**	40.14 ± 18.45	42.79 ± 19.27	36.47 ± 17.02	0.236
Ellipse area (mm^2^)	14.71 ± 20.33	16.27 ± 23.27	11.13 ± 10.55	0.304	10.43 ± 11.79	12.20 ± 13.64	6.82 ± 5.09	0.079	8.84 ± 6.70	8.66 ± 6.32	9.10 ± 7.35	0.821
Principal axis ellipse	5.73 ± 2.9	5.89 ± 3.17	5.37 ± 2.44	0.480	4.54 ± 2.36	4.87 ± 2.61	3.86 ± 1.58	0.102	4.78 ± 1.96	4.72 ± 2.34	4.86 ± 1.96	0.833
Second axis ellipse	2.65 ± 1.6	2.82 ± 1.73	2.25 ± 1.07	0.142	2.37 ± 1.34	2.58 ± 1.47	1.95 ± 0.89	0.072	2.18 ± 0.90	2.17 ± 0.80	2.19 ± 1.03	0.944

Values are presented as mean ± standard deviation or percentages. *t*-test square statistics was applied. Statically significant between body mass status group for each age group are highlighted in bold. C1: left forefoot load; C2: right forefoot load; C3: left hindfoot load; C4: right hindfoot load; F: female; FPI: Foot Posture Index; L: left; LLAS: Lower Limb Assessment Score; M: male; NW: Normal Weight; Ob: Obesity; OW: Overweight; R: Right. ^a^
*p* shows differences for all variables between groups of body mass status, except ^b^ for gender which shows differences in body mass index between sexes.

**Table 2 children-10-00696-t002:** Associations of BMI with type of foot, laxity, foot strength and baropodometric variables by age groups in children.

	5 to 6 Years			7 to 8 Years			9 to 10 Years		
Variables	R^2^	β	*p* Value	R^2^	β	*p* Value	R^2^	β	*p* Value
**Type of foot, laxity and foot strength**									
FPI total (Score)	0.02	0.131	0.247	0.01	0.082	0.505	0.01	−0.050	0.729
Lunge test (°)	0.04	−0.210	0.062	0.05	−0.212	0.085	0.02	−0.129	0.371
Beighton Scale (Score)	0.01	−0.072	0.527	0.06	0.241	**0.049**	0.02	−0.145	0.313
Right LLAS (Score))	0.04	−0.193	0.088	0.00	0.017	0.887	0.02	−0.131	0.364
Left LLAS (Score)	0.03	−0.169	0.137	0.00	0.006	0.963	0.02	−0.127	0.377
Right eversion (N)	0.03	0.160	0.158	0.04	0.206	0.094	0.00	0.016	0.911
Left eversion (N)	0.21	0.460	**<0.001**	0.01	0.083	0.504	0.02	0.126	0.381
Right inversion (N)	0.20	0.450	**<0.001**	0.09	0.298	**0.014**	0.00	0.018	0.904
Left inversion (N)	0.10	0.312	**0.005**	0.06	0.242	**0.048**	0.00	0.050	0.731
Right plantarflexion(N)	0.02	0.396	**<0.001**	0.05	0.218	0.076	0.00	0.012	0.931
Left plantarflexion (N)	0.11	0.329	**0.003**	0.06	0.238	0.052	0.00	0.064	0.656
Right dorsiflexion (N)	0.10	0.313	**0.005**	0.02	0.144	0.244	0.00	−0.008	0.954
Left dorsiflexion (N)	0.02	0.136	0.230	0.01	0.117	0.343	0.00	−0.051	0.725
**Static variables**									
Left–right difference (%)	0.00	0.054	0.638	0.01	0.114	0.360	0.07	0.058	0.058
Forefoot static (%)	0.03	−0.159	0.161	0.02	−0.124	0.317	0.00	−0.045	0.754
Rearfoot static (%)	0.03	0.163	0.150	0.02	0.124	0.317	0.00	0.045	0.754
C1 static (%)	0.01	−0.121	0.289	0.04	−0.197	0.110	0.00	−0.026	0.857
C2 static (%)	0.03	−0.170	0.134	0.00	−0.013	0.916	0.00	−0.047	0.741
C3 static (%)	0.02	0.145	0.203	0.01	−0.085	0.489	0.01	0.075	0.603
C4 static (%)	0.01	0.081	0.478	0.06	0.248	**0.043**	0.00	−0.004	0.976
**Stabilometric variables (static)**									
Left-right difference (%)	0.00	0.002	0.986	0.01	0.100	0.418	0.09	0.303	**0.032**
Front stabilometric (%)	0.02	−0.147	0.194	0.00	0.026	0.832	0.05	−0.213	0.137
Rear stabilometric (%)	0.02	0.129	0.256	0.00	−0.026	0.832	0.05	0.213	0.137
C1 stabilometric (%)	0.01	−0.100	0.378	0.00	−0.053	0.668	0.02	−0.157	0.275
C2 stabilometric (%)	0.02	−0.123	0.280	0.01	0.092	0.454	0.04	−0.199	0.166
C3 stabilometric (%)	0.00	0.046	0.688	0.02	−0.125	0.312	0.04	0.189	0.187
C4 stabilometric (%)	0.02	0.127	0.265	0.01	0.083	0.500	0.00	0.026	0.856
**Stabilometric variables (gravity center)**									
Minimum *x*-axis (mm)	0.02	0.138	0.224	0.00	0.062	0.619	0.01	0.091	0.529
Minimum *y*-axis (mm)	0.02	0.139	0.223	0.01	0.082	0.507	0.00	0.044	0.756
Maximum *x*-axis (mm)	0.01	−0.121	0.287	0.03	−0.165	0.181	0.00	−0.001	0.997
Maximum *y*-axis (mm)	0.00	−0.057	0.618	0.07	−0.263	**0.031**	0.03	0.177	0.218
Interval x (mm)	0.05	−0.233	**0.039**	0.06	−0.240	**0.050**	0.01	−0.113	0.434
Interval y (mm)	0.03	−0.163	0.151	0.09	−0.296	**0.015**	0.02	0.149	0.301
Average x (mm)	0.00	0.037	0.749	0.01	−0.077	0.537	0.01	0.070	0.631
Average y (mm)	0.00	0.049	0.669	0.02	−0.133	0.281	0.02	0.142	0.324
Distance traveled (mm)	0.06	−0.243	**0.030**	0.05	−0.232	0.059	0.03	−0.176	0.221
Ellipse area (1DS) (mm^2^)	0.04	−0.195	0.084	0.06	−0.236	0.054	0.00	0.010	0.947
Principal axis ellipse (mm)	0.04	−0.188	0.096	0.09	−0.291	**0.017**	0.01	0.088	0.541
Secondary axis ellipse (mm)	0.05	−0.231	**0.040**	0.04	−0.198	0.108	0.00	−0.055	0.702

Statically significant are highlighted in bold. C1: left forefoot load; C2: right forefoot load; C3: left hindfoot load; C4: right hindfoot load; FPI: Foot Posture Index; L: left; LLAS: Lower Limb Assessment Score; R: Right.

## Data Availability

Not applicable.

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
