# Peer review of "Overweight and Obesity: Its Impact on Foot Type, Flexibility, Foot Strength, Plantar Pressure and Stability in Children from 5 to 10 Years of Age: Descriptive Observational Study"

_children, 2023, doi:10.3390/children10040696_

Round 1

Reviewer 1 Report

The topic of this research is interesting and related to the wide readership of the journal. This paper can be considered further for publication if the following major revisions are completed.

1. Line 353-356: Please discuss the important clinical implication of this study. How is this data of any use to whoever it may concern. Who does it concern? It is obvious that OW and OB is not good for health.

2. What has this study brought to light which was not previously known about OW and OB in children? Stating this clearly would strengthen his document, and show the authors’ motivation to conduct this study.

3. I would suggest you to improve “2.2.3. Baropodometry” section and discuss recent published articles that focused on the development of novel portable designs to capture the plantar pressures.  Please discuss the following articles and cite it to help the reference and discussion improvement.

i) A Portable Insole for Foot Plantar Pressure Measurement Based on A Pressure Sensitive Etextile and Voltage Feedback Method. https://doi.org/10.1109/ICDSP.2018.8631870

ii) Diabot: Development of a Diabetic Foot Pressure Tracking Device. https://doi.org/10.3390/j6010003

iii) Design of Low Cost Smart Insole for Real Time Measurement of Plantar Pressure. https://doi.org/10.1016/j.protcy.2015.07.020

4. The work is well written and provides good results, which are properly presented in the tables, but their discussion can be deepened. Some discussions are necessary for the introduction to provide the readers with a big picture. Also, numerous minor mistakes in English writing have been found. Please polish the manuscript to avoid errors.

5. The authors can consider the items below for improving the conclusion section: - Restate the research topic in conclusion. - Summarize the main points. - State the significance or results. - Avoid repeating information that you have already discussed. - Mention the model's name, and the advantages and disadvantages of the model. - Mention limitation of the study. - Provide some recommendations for future potential researchers.

6. Additional minor comments that need to be addressed:

      i.         Line 36: First occurance of abbreviation ‘NW’ was not defined. (Normal Weight)

     ii.         Line 35-39: Results sentence need to be rephrased. Information is not clear.

   iii.         Line 58: ‘are/were’ instead of ‘have’

   iv.         Line 79-82: Needs rephrasing.

     v.         Line 181-188: Is this a formatting error?

   vi.         Line 198: I think OW and OB should not be clubbed into one category, as OW is a subset of OB. Refer BMI age chart on https://www.cdc.gov/healthyweight/assessing/bmi/childrens_bmi/about_childrens_bmi.html. Please comment on this.

  vii.         Line 247, 249, 255: ‘any’ instead of ‘none’

viii.         Line 278-280: Needs rephrasing.

   ix.         Line 286-288: “…it could explain why our…”. Remove “our” from this line.

Reviewer 2 Report

Thank you for allowing me to review this manuscript, I found it very interesting. One suggestion I would make is to include the static nature of the data collection as a limitation for this. You mention several time changes in movement related to BMI, but do not have motion data, only static data. We cannot just infer that static positioning will directly impact dynamic movements. I would strengthen that connection and then list this as a limitation for the overall results. 

Reviewer 3 Report

I congratulate the authors for carrying out this research entitled "Overweight and obesity: its impact on foot type, flexibility, foot strength, plantar pressure and stability in children from 5 to 10 years of age. Descriptive observational study".

The results are quite interesting, the study objectives were answered, the results were compared with other results with good discussion. But I have some queries:

- Why did you specifically choose children from 5 to 10 years? Justify this inclusion by quoting references in the introduction.

-  Abbreviate "NW" during first time use both in the abstract and in line 65.

- You measured many variables like Anthropometric measures, Type of foot, laxity and foot strength and Baropodometry. Did you follow any specific time for measurement in all the children? I mean were all the children measured at the same schedule? (For example; after breakfast, after lunch, etc.). Especially during weight measurement. 

- You mentioned in line 242 - "In children 7-8 years, BMI were positively associated with inversion and left dorsiflexion strength (all p<0.05)". But in table 2, the p-value for left dorsiflexion strength is 0.343, not significant.

Round 2

Reviewer 1 Report

The authors have addressed all my comments. Can be accepted with minor english corrections and spell checks.